# Diatoms in Volcanic Soils of Mutnovsky and Gorely Volcanoes (Kamchatka Peninsula, Russia)

**DOI:** 10.3390/microorganisms9091851

**Published:** 2021-08-31

**Authors:** Alfiya Fazlutdinova, Yunir Gabidullin, Rezeda Allaguvatova, Lira Gaysina

**Affiliations:** 1Department of Bioecology and Biological Education, M. Akmullah Bashkir State Pedagogical University, 450008 Ufa, Russia; alfi05@mail.ru; 2Department of Information Systems and Technologies, M. Akmullah Bashkir State Pedagogical University, 450008 Ufa, Russia; junigobi@gmail.com; 3Laboratory of Botany, Federal Scientific Center of the East Asia Terrestrial Biodiversity, 690022 Vladivostok, Russia; allaguvatova@yandex.ru; 4All-Russian Research Institute of Phytopathology, 143050 Odintsovo, Russia

**Keywords:** volcanic eruptions, lifeless substrates, restoration, biodiversity, Southern Kamchatkan subduction zone, Mutnovsky geothermal field, *Eunotia curtagrunowii*, *Humidophila contenta*, *Caloneis bacillum*, *Pinnularia borealis*

## Abstract

Volcanic activity has a great impact on terrestrial ecosystems, including soil algae in general and diatoms in particular. To understand the influence of volcanoes on the biodiversity of diatoms, it is necessary to explore the flora of these microorganisms in regions with high volcanic activity, which includes the Kamchatka peninsula. During the study on diatoms in the soils of Mutnovsky and Gorely volcanoes of Kamchatka, 38 taxa were found. The Mutnovsky volcano diatom flora was more diverse and accounted for 35 taxa. *Eunotia curtagrunowii*, *Humidophila contenta*, and *Pinnularia borealis* were the dominant species. In the Gorely volcano, only 9 species were identified, with *Caloneis bacillum* and *Pinnularia borealis* prevailing in the samples. Overall, the genera *Pinnularia* and *Eunotia* were the most diverse in the studied area. The diatom flora of the studied volcanoes comprises mostly cosmopolitan small-sized taxa with a wide range of ecological plasticity. Our data confirm the high adaptive potential of diatom algae and add new knowledge about the ecology and biogeography of this group of microorganisms.

## 1. Introduction

Volcanoes, microorganisms, and water are important factors supporting life on our planet. Fossil records confirm that volcanic glass serves as a bioreactor for generating microbial life [1]. Volcanic activity has a great impact on the development of terrestrial ecosystems due to the influence of various chemical elements with volcanic emissions, changes in the temperature regime, transformations of existing landscapes, and the creation of new environment [2].

Algae (including diatoms) are among the first microorganisms that settle on lifeless substrates after a volcanic eruption and initiate the succession during their overgrowth [3,4].

The study of the colonization of volcanic substrates by algae has been associated with the largest volcanic eruptions [5]. These studies started in 1886 when the eruption of the Krakatoa volcano led to the immersion of most of the island underwater and the “sterilization” of the rest of its surface. Cyanobacteria were the first organisms that settled on the volcanic surface, and they played the most important role in the colonization of the lifeless volcanic substrate [6].

The next stage in the study of the algal flora of volcanic emissions began after the eruption of an underwater volcano near the coast of Iceland and the emergence of Surtsey Island in 1963 [4,7,8,9,10,11]. At the initial stages of the appearance of the vegetation currently covering this island, diatoms and green algae prevailed in the algal flora. In total, 71 species of diatoms, including numerous aquatic forms, were identified [12]. Cyanobacteria and yellow-green and euglenoid algae were much fewer in number.

A well-studied example of the colonization and succession of cyanobacteria and algae in another volcanically active region—the Hawaiian Islands—was described in the literature. A positive correlation between the diversity and abundance of algae with the level of nutrients and the accumulation of organic matter was revealed here [13].

In the study of the communities of soil algae inhabiting andosols with volcanic glasses on Deception Island [14,15], cyanobacteria and diatoms were dominant in the algal communities.

Volcanic ashes are low in organic nutrients and energy sources; therefore, only microorganisms with an economical metabolism and specific adaptations to this environment can survive [16,17]. Different types of soils can be formed from volcanic ash depending on the characteristics of the soil-forming process in a particular habitat [18].

Green algae form a mucilage, stabilizing the fine-grained volcanic soil. In addition, the mucilage protects them from water fluctuations and defends the cells from loss of water [19]. It has been shown that diatoms precipitate silicon and calcite, which can also contribute to the aggregation of mineral particles [20].

Kamchatka is part of the Kuril–Kamchatka island (or volcanic) arc. It is one of the most seismically active regions on the Earth, with about 30 active and potentially active volcanoes in the Kuril–Kamchatka region [21,22,23]. As a result of the eruptions of these volcanoes, millions of tons of volcanic ash are emitted into the atmosphere [24].

Information on the algae of Kamchatka’s volcano soils is limited and based on a sporadic sampling. The numerous hot springs associated with volcanic activity have traditionally attracted a great deal of attention from researchers [25,26,27,28]. At the same time, there is evidence that volcanic soils can harbor a significant diversity of algae and cyanobacteria [5,29]. In soils of the Kuril–Kamchatka group of volcanoes—Tyatya, Golovin, and Mendeleev on Kunashir Island and Tolbachik in Kamchatka—74 species of algae were found. A few species of algae were also reported from the lava caves of the Gorely volcano [30]. As stated before, diatoms play an important part in the primary succession of volcanic substrates, but the data about the biodiversity of diatoms of volcano soils, including Kamchatka soils, are very limited. 

Our study aimed to investigate the diversity of diatoms in volcano soils of Mutnovsky and Gorely volcanoes, which are some of the active volcanoes of Kamchatka [31,32] and can be considered as model objects for observing the colonization of lifeless volcanic substrates by microorganisms.

## 2. Materials and Methods

### 2.1. Studied Area

The Kamchatkan peninsula is one of the most active volcanic regions on the Earth [33] (Figure 1). The geochemistry of the Kamchatka volcanic rocks is typical of global arc volcanism. They contain a high amount of large ion lithophile elements such as Rb, Ba, K, Pb, and Sr in comparison with high field strength elements such as Nb, Zr, and Ti [34,35,36,37]. 

Mutnovsky and Gorely volcanoes belong to the Mutnovsky geothermal field, which is part of the Mutnovsky geothermal region with an area of about 750 km^2^ [38] (Figure 1). The territory of the geothermal field is a volcanic plateau at 700–900 m a.s.l. elevation divided by the Falshivaya, Mutnovskaya, and Zhirovaya rivers. The Mutnovsky geothermal field is characterized by numerous thermal water and steam discharges, scoria cones, and extrusive bodies, together with hydrothermally altered rocks [38,39,40]. The composition of volcanic rocks in the region includes tuff and lava of andesite, andesidacite, and andesibasalt with inclusions of subvolcanic bodies and dikes of variable composition (diorite, dioritic porphyrite, basalt) [38,40,41] (Table 1).

Due to the open position and windwardness of both slopes, the area has similar weather conditions on the eastern and western slopes of the volcanic highlands [42]. The eastern slopes receive the maximum precipitation for the whole of Kamchatka, at 2500 mm per year. High humidity, heavy rainfall, and frequent hurricane winds create extremely unfavorable weather conditions in the area. Because of the small width of the peninsula, the influence of the Pacific Ocean is great [43].

The studied area is located in the southeastern soil province [44], which belongs to the zone of stone birch forests. The volcanic soils of Kamchatka are distinctive for the very low values of the potential buffering capacity for potassium [45] and the increased content of copper, manganese, scandium, vanadium, and silver against the background of low concentrations of other microelements [46]. The surface organogenic horizon within the province was formed on andesite–basaltic gray coarse-grained ash of the eruption of the Ksudach volcano in 1907. All ash horizons, except for the near-surface horizon, are predominantly acidic. The humus content in organogenic horizons is high, at up to 5–10%. The reaction of the environment in soils is acidic and slightly acidic. The degree of saturation with bases is low: on average, 8–11%. The ocher illuvial horizons are characterized by the accumulation of soluble forms of iron and aluminum up to 3% and 8%. In the elfin belt at an altitude of 700–1000 m, there are humus–ocher soils (under alder forests), peaty illuvial–humus soils, and peaty illuvial–humus volcanic soils (under cedar forests). The soils of the subalpine elfin zone are characterized by the peaty character of the modern organogenic horizon, the humus or one and a half-turfy nature of the buried organogenic horizons, the predominance of brown tones in the color of the illuvial–metamorphic horizons, the retardation (in comparison with the forest zone) of the processes of weathering and the accumulation of amorphous substances, high acidity, and unsaturation [47].

The vegetation cover is characterized by a significant decrease in the altitudinal vegetation belts [48]. The vegetation cover of the studied volcanoes is dominated by elite forests and mountain tundra. The surfaces of loose volcanic deposits are almost lacking vegetation.

The Mutnovsky volcano is situated 70 km southwest of Petropavlovsk-Kamchatsky on the Kamchatka Peninsula. Mutnovsky belongs to the Eastern Volcanic Front—one of the three major volcanic chains in Kamchatka, which were created by the northwest-directed subduction of the Pacific Plate [49]. The Mutnovsky volcano is an elongated massif consisting of four merged cones. The explosions, which took place in several stages, first formed two huge craters in the southwest and northeast, resembling a figure eight and stretching for 4 km. The merged crater of the Mutnovsky volcano with a height of 2323 m is the largest among the craters of active volcanoes in Kamchatka. The diameter of the craters reaches 1.5–2 km, and the depth is 400 m. Subsequent explosions in the southwest crater formed a deep funnel, previously occupied by a lake and now by a glacier. In addition, the active funnel is superimposed on the northern edge of the southwest crater, which is a closed bowl with a depth of about 150 m, with steep walls and a flat bottom that is 100 × 150 m in diameter. The Vulkannaya River, cutting through the bottom and western walls of the northern crater, forms a powerful 80 m waterfall at the exit, and below, it forms a deep canyon—the Opasny Ravine [50].

In the craters of the volcano and on its northern slope, there are high-temperature fumaroles; steam-gas jets, the outlets of which are framed by volcanic sulfur; and thermal springs in the form of water and mud pots. The most intense fumarolic activity is concentrated mainly in the northeast crater and the active funnel. In the first crater, three groups of steam-gas outlets are constantly operating—this is the upper fumarole field, with fumarole temperatures of more than 300 °C—and two relatively isolated groups on the Donnoye fumarole field are present with outlet temperatures up to 150 °C. In the active funnel, the most powerful gas activity is concentrated on the southwestern wall [50].

The production of mostly low K basaltic lavas with rare earth element patterns is a unique feature of Mutnovsky. This feature is similar to the classic island arc tholeiites [51]. Mutnovsky lavas belong to a tholeiitic igneous series with 48–70% SiO_2_. Basalts and basaltic andesites have relatively low K_2_O and Na_2_O and high FeO* and Al_2_O_3_ in comparison with other Kamchatka volcanic rocks [49].

The Gorely volcano is situated 75 km southwest of Petropavlovsk-Kamchatsky. It has a height of 1800 m [52]. The Gorely volcano consists of two large parts: an ancient shield-like structure, crowned with a 13 km caldera, and a modern structure like a complex stratovolcano. The modern structure covers an area of 150 km^2^, located in the center of the caldera, and is composed mainly of basaltic and andesite–basaltic lavas. The structure itself and the morphology of the streams resemble the Hawaiian type of volcanic manifestation. Its top is framed by a chain of explosive craters, and about 30 flank cones are noted on the slopes [50,53,54,55]. The ancient part of the volcano covers an area of 650 km. Its gentle slopes can be traced in the north to the upper reaches of the river Paratunka, northeast to the headwaters of the river Zhirovaya, in the east to the Falshivaya and Vulkannaya rivers, in the south to the northern slopes of the Asach volcano, and in the west to Tolmachev Dol. The structure is composed of andesite–basalt and andesite lavas and mainly pyroclastics of andesite–dacite and dacite composition. Lavas and extrusions of liparite rhyolites are found here in small quantities. Of greatest interest are pyroclastic formations such as sintered tuffs and ignimbrites. In the sections of the walls, cutting through the edifice, one can trace here the transformation of pyroclastic material from loose pumice to sintered tuffs and ignimbrites, as well as tuff lava [50].

**Table 1 microorganisms-09-01851-t001:** Comparisons of the geochemistry of Mutnovsky and Gorely volcanoes [31,47,49,56,57,58,59].

	Mutnovsky Volcano	Gorely Volcano
Volcanic composition and chemical content of the rocks	Basalts and basaltic andesites with relatively low K_2_O and Na_2_O and high FeO* and Al_2_O_3_ two-peroxene dacites, composed mainly of low-potassium and calc-alkaline basalts. In terms of SiO_2_–K_2_O relations, they belong to the low- and moderate-potassium varieties of the calc-alkaline series, plotting along with the tholeiitic–calc-alkaline series; in terms of the alumina index, they are moderately aluminous rocks. The young basalts are enriched in MgO and CaO but differ in low contents of SiO_2_, TiO_2_, Al_2_O_3_, and Na_2_O.	Basalts and andesitodacites with low K_2_O, high contents of CaO, TiO_2_, and total iron, with a predominance of intermediate basaltic andesite rocks. All varieties of young volcanic rocks have elevated K_2_O contents and correspond to the high potassium calc-alkaline series, with normal alkalinity. Some lavas of the youngest eruptions have elevated alkali contents and correspond to the subalkaline series. Most lavas of the riftogenic zone are clustered around the dividing line of the calc-alkaline–tholeiitic series.
pH of constituent rocks	10–15% of acid-medium rocks	Acid andesites
Last eruption	March 2000	Summer 2010
Concentration of SiO_2_	48–70%	51–57%
Tephra	Gravel and lapilli of dense andesite with interlayers of yellowish silty sands, thin ashes with the inclusion of larger grains of sand and gravel	Black-gray volcanic sand and slag

### 2.2. Sample Collection

Volcanic soil samples were collected in August 2010 from 13 sites (Table 2, Figure 2). Samples (100 g of soil) were taken with metal cylinders, according to the terrestrial diatom sampling method [60], and put into sterile paper bags. The samples were examined according to previously described methods [28]. For diatom cell cleaning, 1 g of each soil sample was diluted with 10 mL of deionized water and 10 mL of nitric acid. For water evaporation, the mixture was boiled to a double volume reduction and, after cooling to room temperature, was washed four times using deionized water.

Diatom cells were settled by sedimentation during the washing procedure. Then, suspensions of the washed cells were dried on glass coverslips and mounted on permanent slides with Naphrax following standard methods [61]. Diatoms were examined at 1000× magnification, using Zeiss Axio Imager A2 light microscopes (Carl Zeiss, Jena, Germany), equipped with oil immersion objectives with differential interference contrast (DIC) and Axio Cam MRc cameras. Permanent slides were deposited in the Bashkortostan Collection of Algae and Cyanobacteria (BCAC) (WDCM 1023, Ufa, Russia). The identification of species was conducted using relevant references [62,63,64,65,66] and recent publications [67,68,69,70]. For descriptions of the ecology of species, standard flora and related literature data were used [71,72,73,74,75,76,77,78,79].

The floristic similarity of volcanoes was accessed with the Sørensen–Czekanowski coefficient [28].

For the comparison and visualization of the similarities between diatom communities from Mutnovsky and Gorely volcanoes, a Venn diagram was used [80,81]. The Venn diagram was created with the InteractiVenn tool [82].

The abundance of the diatom species was estimated with the previously described method [28,83]. According to this method, the minimum abundance of algae was 1 point (1–15 on a slide) and the maximum abundance was 15 points (more than 50 on a slide). Species with an abundance of 13–15 points were dominant in the sample.

## 3. Results

In volcanic soils of Mutnovsky and Gorely volcanoes, 38 taxa of diatoms from 20 genera, such as *Adlafia* Gerd Moser, Lange-Bertalot and Metzeltin, *Caloneis* Cleve, *Diatoma* Bory, *Eunotia* Ehrenberg, *Fragilariforma* D.M. Williams and Round, *Gomphonema* Ehrenberg, *Hantzschia* Grunow, *Humidophila* (Lange-Bertalot and Werum) R.L.Lowe and al., *Luticola* D.G.Mann, *Muelleria* (Frenguelli) Frenguelli, *Navicula* Bory, *Nitzschia* Hass, *Pinnularia* Ehrenberg, *Planothidium* Round and L.Bukhtiyarova, *Platessa* Lange-Bertalot, *Psammothidium* L.Buhtkiyarova and Round, *Sellaphora* Mereschowsky, *Stauroneis* Ehrenberg, *Staurosirella* D.M.Wiliams and Round, and *Tabullaria* (Kützing) D.M.Williams and Round were identified (Figure 3A–W, Table 3). The genera *Pinnularia* (seven species), *Eunotia* (five species), and *Caloneis* (three species) were the richest in terms of the number species and were recorded in almost all the samples studied. On average, we found 6.6 species per sample, but the number of species increased with the distance from the volcanic craters. In the samples K6 on the trail along the edge of the crater, K7 down the east slope, and K8 at the edge of a crater with a lake from the Gorely volcano, diatoms were not found. In the samples K12, K13, and K11, three, one, and seven taxa were detected, respectively. Similarly, on the Mutnovsky volcano, in sample K3 300 m from the top of the volcano, 5 diatom species were identified; in K1 in the bushes, K2 in alder forest, and K4 in the Vulkannaya River canyon, 13, 22, and 9 taxa were found, respectively (Table 2 and Table 3). 

Soils of the Mutnovsky volcano were more diverse in diatoms in comparison with those of the Gorely volcano. In samples from the Mutnovsky volcano, 35 taxa (94.6%) were identified, and only 9 taxa (24.3%) were found at the Gorely volcano (Table 2, Figure 4). *Pinnularia borealis* (Figure 3N) were dominant in the soils of both studied volcanoes. At the Mutnovsky volcano, *Eunotia curtagrunowii* (Figure 3F) and *Humidophila contenta* (Figure 3J) dominated; in soils of the Gorely volcano, *Caloneis bacillum* (Figure 3C) prevailed. Only six taxa—*Caloneis bacillum, Eunotia curtagrunowii, Pinnularia borealis, P. intermedia, P. microstauron*, *P.* cf. *subcapitata*—were found in both volcano soils (Table 3, Figure 4). Therefore, the floristic similarity between diatom floras of Mutnovsky and Gorely volcanic soils according to the Sørensen–Czekanowski coefficient was very low, at only 27.9%. 

The diatoms in the studied habitats are adapted to living conditions not only in terrestrial but also in aquatic ecosystems and include bottom, littoral, and epiphytic species, as well as transitional forms: littoral–epiphytic, plankton–bottom, and bottom–epiphytic (Table A1). In total, 21 taxa were benthic (56.7%). Transitional forms of benthic forms (plankton–benthic and benthic–epiphytic) were found for five species (13.5%). Epiphytic forms were represented by seven species (18.9%), while littoral was represented by one species *Staurosirella pinnata*. The littoral–epiphytic group included four species (10.8%): *Eunotia bilunaris*, *E. curtagrunowii*, *Fragilariforma virescens* var. *exigua*, and *Tabularia fasciculata* (Figure 3W). Bottom–epiphytic forms were also represented by four species (10.8%): *Diatoma vulgaris*, *Humidophila contenta*, *Luticola mutica* (Figure 3K), and *Navicula cincta*. Plankton–bottom species were represented by only one representative of diatoms: *Diatoma tenuis*.

Concerning mineralization, four ecological groups were found: indifferent, halophilic, halophobic, and mesagolob diatoms (Table A1). The largest group (22 species, 59.4%) was represented by indifferent species. The halophobic group included eight species (21.6%): *Caloneis dubia* (Figure 3D), *Eunotia bilunaris*, *E. curtagrunowii*, *E. fallax*, *E. paludosa*, *E. sudetica* f. *minor*, *Pinnularia* cf. *subcapitata* (Figure 3R), and *Platessa oblongella* (Table A1). Halophilic taxa were represented by seven species (19%) inhabiting slightly salted substrates: *Adlafia aquaeductae* (Figure 3B), *Diatoma tenuis*, *Luticola mutica*, *Navicula cincta*, *Sellaphora mutata*, *S. pupula*, and *Staurosirella pinnata*. One species of diatoms (2.7%) belonging to the mesohalobes—*Tabularia fasciculata*—was found.

Despite the fact that the studied volcanic soils were acidic and slightly acidic, a predominance of taxa that were indifferent to pH was detected (Table A1). In total, 15 indifferent species (40.5%), such as *Adlafia aquaeductae*, *Fragilariforma virescens* var. *exigua*, *Gomphonema parvulum* (Figure 3H), *Pinnularia intermedia* (Figure 3O), *Pinnularia microstauron*, and several representatives of the genus *Caloneis* were detected. Alkaliphiles accounted for 37.3% of the total number of taxa, including *Caloneis lancettula* (Figure 3E), *Diatoma tenuis*, *Hantzschia amphioxys* (Figure 3I), *Humidophila contenta*, *Luticola mutica*, *Nitzschia palea* (Figure 3M), *Pinnularia intermedia*, *Staurosirella pinnata*, and others. Acidophilic taxa were represented by *Eunotia bilunaris*, *E. curtagrunowii*, *E. fallax* (Figure 3G), *E. paludosa*, *E. sudetica* f. *minor*, *Pinnularia borealis*, and *Pinnularia* cf. *subcapitata*. Two alkalibionts, *Nitzschia* cf. *ovalis* (Figure 3L) and *Diatoma vulgaris*, were also noted.

The geographic structure of the diatom communities of the studied volcanic soils also has its peculiarities. The most widespread communities were the cosmopolitan species (31 species and intraspecific taxa), and the arcto-alpine species were represented by 5 taxa (*Eunotia curtagrunowii*, *E. paludosa*, *E. sudetica* f. *minor*, *Fragilariforma virescens* var. *exigua*, *Pinnularia intermedia*). Boreal species were represented by only two species: *Psammothidium ventrale* (Figure 3U) and *Sellaphora submuralis* (Table A1).

## 4. Discussion

We revealed an increase in the number of diatom species with the distance from the volcano craters and with the appearance of primary soil and higher plant vegetation (Table 2 and Table 3). Therefore, the studied samples of volcanic substrates located closer to the crater were either sterile or had a poor species diversity; for example, samples K6, K7, K8, K9, K10 (Table 2 and Table 3). In the samples K1, K2, K 9, and K11, an increased number of species was observed (Table 3). It is necessary to note that diatom algae are important for the colonization of areas of the Earth’s surface after volcanic eruptions [4]. The volcano represents the model of restoration of bare substrates and soil-forming processes [84], where the crater of the volcano is the first stage of succession. The positive interactions between high plants and microorganisms have been discussed in several publications. Plants positively influence microbial communities and carbon accumulation in soils [84,85]. In turn, many groups of prokaryotic [84,86,87,88,89,90,91] and eukaryotic microorganisms [92,93,94], including diatom algae, are very important in soil formation and functioning. 

Soils at the Gorely volcano were particularly poor regarding diatoms, with only nine taxa found here (Table 3, Figure 4). This may be a consequence of several factors. The samples were collected from the Gorely volcano in August 2010 during eruptive activity in sites K12 and K13 (Table 2). Possibly, the toxicity of the volcanic soils of the Gorely volcano were much higher after the eruption, which also contributed to the small number of diatom species in the soil.

Interestingly, only three taxa were specific for the Gorely volcano soils (Table 3, Figure 4): *Caloneis lancettula*, *Gomphonella parvulum*, and *Pinnularia* sp.3. Possibly, these species are very resistant and can establish the first stage of succession during the restoration of volcanic substrates. In the last decade, new data about the ecology of many terrestrial diatoms have been discussed [70,79,95,96]. It should be stressed that our knowledge about the ecological behavior of terrestrial diatom taxa is very limited. For example, in freshwater habitats, *Caloneis lancettula* was reported as a very sensitive taxon, but it was abundant in disturbed soils in the Attert River basin in Luxembourg [79]. Our study confirms that this species is very adaptive to extreme environmental conditions in terrestrial habitats.

As mentioned above, at the Mutnovsky volcano, the processes of ecosystem restoration, expressed as an increase in the soil fertility and the appearance of bushes and alder forests, were observed. Therefore, in the soils of the Mutnovsky volcano, a greater number of species (29) was found in comparison with the Gorely volcano. At site K2, the maximal number of taxa (22) was detected. The high number of diatoms found here was likely due to the formation of “true” soil, more stable soil humidity (Table 2), interactions with high plants, and other factors, influenced by the late stage of volcanic substrate restoration [84].

The formation of the species composition of diatoms is probably influenced by the accidental transfer of cells from nearby regions. For example, *Navicula* sp. and *Nitzschia* sp. were the most common airborne algae [97]. 

The local selection of species to the physicochemical conditions of volcano soils is also very important. The pH of the soil influenced species abundance. Basically, samples K1–K4 with alkaline soils were more diverse in terms of diatoms in comparison with the samples with acidic soil; for example, K8–K10, K12, and K13. Most organisms live at a pH of 4 to 9, and their optimal growth is observed in an environment close to neutral. At the same time, diatoms are tolerant of extreme pH [79]. The pH has an indirect effect through the solubility of various substances and their accessibility to algae. With an increase in the acidity of the environment, there is a difficulty in the flow of nitrogen, phosphorus, and other mineral elements into the diatom cells. This explains the abundant development of diatoms in neutral and alkaline soils. The acidic reaction of the environment also negatively affects the morphology of diatoms. Perhaps this is the reason why the diversity and abundance of diatoms in soil samples with alkaline volcanic soils increased.

The geology of a volcanic substrate can also influence the species compositions of diatoms [70]. Volcanic rocks of Mutnovsky and Gorely are rich in SiO_2_ and other elements, such as Al, Fe, Si, and Mg (Table 1). Algae contribute to the dissolving of silicate minerals by aiding in the retention of water and acidification by excreting carbonic acid [98]. As a result of exposure to physicochemical and biological factors, insoluble minerals transform into soluble forms. These inorganic nutrients are very important for physiology and biochemistry and promote the survival of diatoms [99,100,101,102,103]. High concentrations of nutrients can positively influence the diatoms on the volcanic substrate at the early stage of the colonization of lifeless substrates after an eruption. On another side, in the primary stages of succession on volcanic substrate, diatoms lack enough moisture and organic nutrients. This circumstance prevents their abundant development. 

We observed a relatively low diversity of diatoms in Mutnovsky and Gorely volcano soils in comparison with other types of soils [79,104,105,106]. This phenomenon could be explained by the very low density, high hydrophilicity, and high filtration capacity of volcanic soils. Such soils resemble a “sponge” through which water easily seeps without lingering in it. Diatoms are called “genuine soil-building algae” and can be underestimated due to their invisibility on the surface [12]. 

It is important to note that, despite the prevalence of taxa indifferent to pH in total, the heterogeneity of ecological characteristics of species in different sites was observed. For the site K2 at the base of the Mutnovsky volcano, with alder forest and humus–ocher soils (Table 2, Table 3 and Table A1), acidophilic taxa *Eunotia curtagrunowii* and *Pinnularia borealis* together with alkaliphilic *Humidophila contenta* were dominant. The frequent occurrence of the genus *Humidophila* in terrestrial ecosystems characterized by insufficient moisture possibly is a consequence of the morphology of the taxon. The reduced external openings of the *Humidophila* frustules allow a reduction in the moisture loss [70].

Some taxa were recorded before in Kamchatka volcanic soil. *Pinnularia borealis* and *Hantzscia amphioxys* were detected in the volcanic soil of the Golovina volcano in Kamchatka [5]. *Hantzschia amphioxys*, *Humidophila contenta*, and *Pinnularia subcapitata* were recorded in volcano soils of the Kuril Islands, which also belong to the Kuril–Kamchatka Island Arc [107]. The genus *Humidophila* has been recorded in freshwater habitats on the islands of Hawai’i, known for their volcanic activity [108]. Representatives of the genus *Navicula* were detected in the outermost part of the Kilau Iki volcano in soil substrate in the periphery of the cinder cone without much destruction during the eruption and in the transitional zone with destroyed vegetation [13]. Several species from Mutnovsky and Gorely volcano soils were recorded in soils of the Malki, Upper Paratunka, and Dachnie thermal springs: *Caloneis lancettula*, *Eunotia curtagrunowii*, *Navicula cincta*, *Pinnularia borealis*, *P. microstauron*, *P.* cf. *subcapitata*, and *Planothidium lanceolatum* [28]. This species might be considered as a typical Mutnovsky geothermal field diatom community.

In Mutnovsky and Gorely volcano soils, we found mostly small-sized species, such as *Caloneis bacillum*, *Humidophila contenta*, and *Eunotia curtagrunowii* (Figure 3). The results obtained once again confirm the hypothesis that small diatoms dominate in aerial communities [95]. 

A comparison of the diatom communities of volcano soils of Kamchatka with volcano soil of other regions reveals interesting peculiarities. In algal communities on an Antarctic active volcano on Deception Island, South Shetlands, diatoms were dominant together with cyanobacteria [15]. The species composition of diatoms in this region has some similarities with the diatoms of Mutnovsky and Gorely volcano soils; for example, the wide distribution of *Pinnularia* genus and presence of *Hantzschia amphioxys*, *Luticola mutica*, *Nitzschia*, and *Muelleria* genera. A wide distribution of cosmopolitan (“ubiquitous”) species with a high range of ecological plasticity, discussed in previous studies [5,15], might be a specific feature of volcanic soils. 

However, there were significant differences in Kamchatka’s investigated volcano soil diatoms communities: the presence of genera *Eunotia* and *Caloneis* with the domination of some species from these genera; the prevalence of *Humidophila contenta*; the existence of genera *Diatoma* and *Sellaphora;* and some cosmopolitan species, such as *Adlafia aquaeductae*, *Gomphonema parvulum*, *Planothidium lanceolatum* (Figure 3S), and others. 

It should be noted that many diatoms in our study, especially cosmopolitan diatoms, could be new taxa or cryptic species. Cosmopolitan and wide-distributed taxa, such as *Pinnularia borealis, Hantzschia amphioxys*, and *Humidophila contenta*, are cryptic or pseudo-cryptic [96,109]. To understand the biodiversity of this unique habitat better, further research using electron microscopy and molecular genetic tools is needed.

Thus, the diatom flora of Mutnovsky and Gorely volcano soils in the Kamchatka peninsula includes mostly cosmopolitan small-sized taxa that are adapted to survival in conditions of the toxic substrate with limited moisture and organic matter. These data once again confirm the high adaptive potential of diatom algae and add new knowledge about the ecology and biogeography of this group of microorganisms.

## Figures and Tables

**Figure 1 microorganisms-09-01851-f001:**
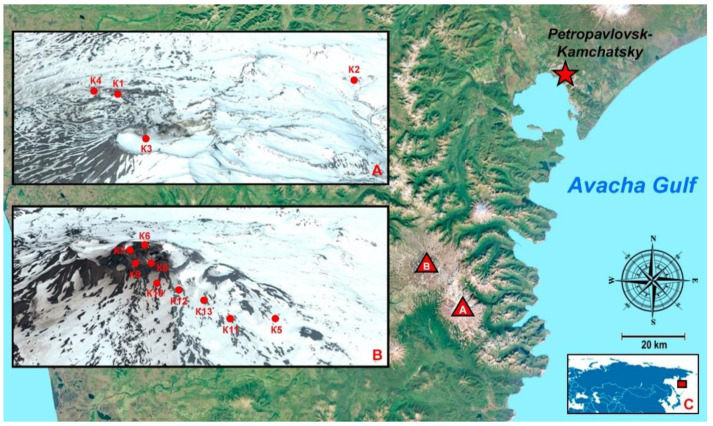
Study area. The red star indicates Petropavlovsk-Kamchatsky city; red triangle with letter (**A**)—Gorely volcano, red triangle with letter (**B**)—Mutnovsky volcano; (**A**)—Mutnovsky volcano, red dots indicate sites K1–K4; (**B**)—Gorely volcano, red dots indicate sites K5–K13; (**C**)—red rectangle indicates the Kamchatka peninsula.

**Figure 2 microorganisms-09-01851-f002:**
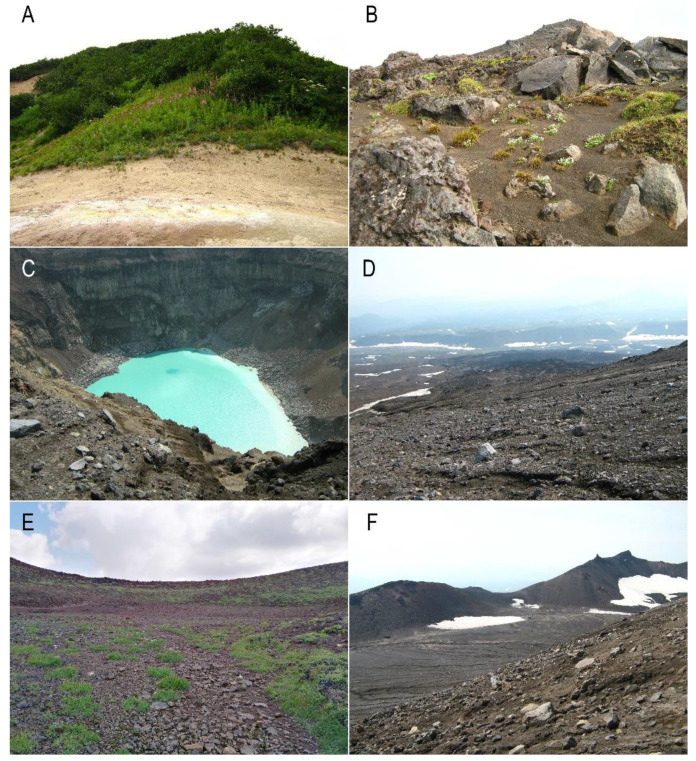
Study sites: (**A**) K2, alder forest at the base of the Mutnovsky volcano; (**B**) K5, flat area among sedges near the Gorely volcano; (**C**) K6, trail along the edge of the Gorely crater; (**D**) K10, down the east slope of the volcano; (**E**) K11, 1000 m from the top of the Gorely volcano; (**F**) K13, 800 m from the top of the volcano.

**Figure 3 microorganisms-09-01851-f003:**
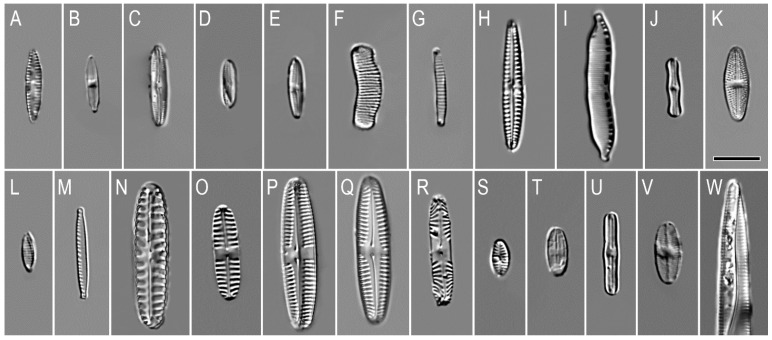
Species of diatoms from Mutnovsky and Gorely volcano soils: (**A**) *Achnanthes linearis* var. *pusilla*; (**B**) *Adlafia aquaeductae*; (**C**) *Caloneis bacillum*; (**D**) *Caloneis dubia*; (**E**) *Caloneis lancettula*; (**F**) *Eunotia curtagrunowii*; (**G**) *Eunotia fallax*; (**H**) *Gomphonema parvulum*; (**I**) *Hantzschia amphioxys*; (**J**) *Humidophila contenta*; (**K**) *Luticola mutica*; (**L**) *Nitzschia* cf. *ovalis*; (**M**) *Nitzschia palea*; (**N**) *Pinnularia borealis*; (**O**) *Pinnularia intermedia*; (**P**) *Pinnularia* sp.1; (**Q**) *Pinnularia* sp.2; (**R**) *Pinnularia* cf. *subcapitata;* (**S**) *Planothidium lanceolatum*; (**T**) *Platessa oblongella*; (**U**) *Psammothidium ventrale*; (**V**) *Sellaphora bacillum*; (**W**) *Tabularia fasciculata*.

**Figure 4 microorganisms-09-01851-f004:**
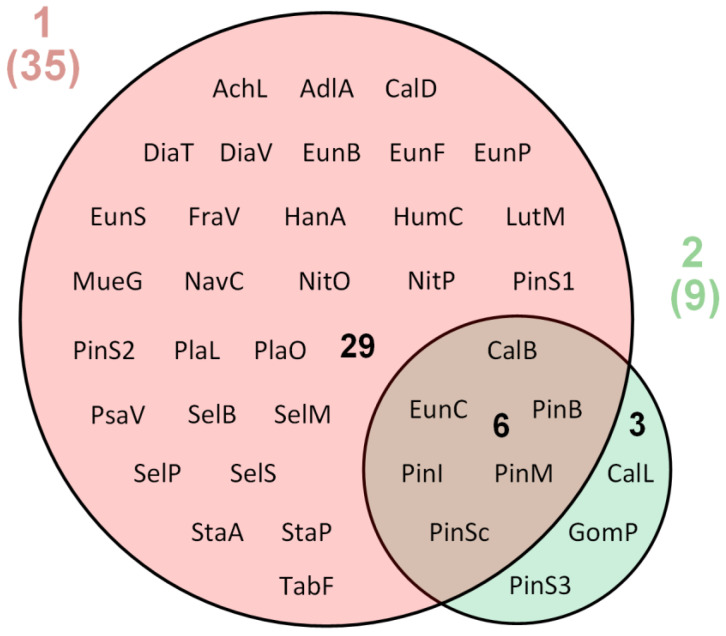
The similarities between diatom communities from the volcano soil studied. 1 (pink circle)—Mutnovsky volcano soil community, 2 (green circle)—Gorely volcano soil community. Numbers in brackets indicates the numbers of species in communities. Species abbreviations: AchL—*Achnanthes linearis* var. *pusilla*; AdlA—*Adlafia aquaeductae*; CalB—*Caloneis bacillum*; CalL—*C. lancettula*; CalD—*C. dubia*; DiaT—*Diatoma tenuis*; DiaV—*D. vulgaris*; EunB—*Eunotia bilunaris*; EunC—E*. curtagrunowii*; EunF—*E. fallax*; EP—*E. paludosa*; EunC—*E. sudetica* f. *minor*; FraV—*Fragilariforma virescens* var. *exigua*; GomP—*Gomphonella parvulum*; HanA—*Hantzschia amphioxys*; HumC—*Humidophila contenta*; LutM—*Luticola mutica*; MueG—*Muelleria gibbula*; NavC—*Navicula cincta*; NitO—*Nitzschia* cf.*ovalis*; NitP—*N. palea*; PinB—*Pinnularia borealis*; PinI—*P. intermedia*; PinM—*P. microstauron*; PinSc—*P.* cf. *subcapitata*; PinS1—*Pinnularia* sp.1; PinS2—*Pinnularia* sp.2; PinS3—*Pinnularia* sp.3; PlaL—*Planothidium lanceolatum*; PlaO—*Platessa oblongella*; PsaV—*Psammothidium ventrale*; SelB—*Sellaphora bacillum*; SelM—*S. mutata*; SelP—*S. pupula*; SelS—*S. submuralis*; StaA—*Stauroneis anceps*; StaP—*Staurosirella pinnata*; TabF—*Tabularia fasciculata*.

**Table 2 microorganisms-09-01851-t002:** Sampling sites.

Number	Description	Name	GPS *	Area	pH	Humidity, %	Type of Soil
1	Canyon of the Vulkannaya river, under the bushes	K1	52°28′29.4″ N 158°06′47.8″ E	M **	9.1	50–68	Mountain–tundra illuvial–humus soils
2	At the base of the volcano, not far from Dachnye springs, alder forest	K2	52°31′54.6″ N 158°11′55.0″ E	M	8.9	75–85	Humus–ocher soils
3	300 m from the top of the volcano	K3	52°27′26.4″ N 158°09′50.4″ E	M	9.1	40–50	Stone talus and placers, rocks
4	In the lower part of the Vulkannaya River canyon	K4	52°28′17.3″ N 158°06′02.4″ E	M	9.1	50–58	Rocks
5	Slope, flat area among sedges	K5	52°32′35.0″ N 158°03′58.2″ E	G ***	5.8	60–70	Illuvial–humus volcanic destructive soils
6	The trail along the edge of the crater, green layer on the surface of the ground	K6	52°33′26.4″ N 158°02′09.2″ E	G	9.0	50–65	Volcanic ash, sand
7	Down the east slope	K7	52°33′19.1″ N 158°01′57.4″ E	G	9.2	55–65	Tundra volcanic illuvial–humus soils
8	At the edge of a crater with a lake	K8	52°33′12.8″ N 158°02′20.7″ E	G	5.0–6.0	40–50	Sulfur deposits around the crater
9	Down the east slope	K9	52°33′10.8″ N 158°02′06.0″ E	G	5.0–6.5	55–65	Tundra volcanic illuvial–humus soils
10	Down the east slope	K10	52°32′53.7″ N 158°02′21.6″ E	G	5.2–6.3	55–65	Tundra volcanic illuvial–humus soils
11	1000 m from the top of the volcano	K11	52°32′27.8″ N 158°03′22.0″ E	G	5.2–6.3	55–65	Tundra volcanic illuvial–humus soils
12	An active crater during the sampling, 500 m from the top of the volcano	K12	52°32′46.2″ N 158°02′39.9″ E	G	8.4–9.0	-	Volcanic ash, sand
13	800 m from the top of the volcano, rare vegetation	K13	52°32′38.7″ N 158°03′03.1″ E	G	5.2–6.3	65–70	Tundra volcanic illuvial–humus soils

Notes. * positions of the sites in the map indicated on Figure 1; M **—Mutnovsky Volcano; G ***—Gorely Volcano.

**Table 3 microorganisms-09-01851-t003:** Diatom taxa recorded at each sampling site.

Taxa	Mutnovsky Volcano				Gorely Volcano								
	K1	K2	K3	K4	K5	K6	K7	K8	K9	K10	K11	K12	K13
*Achnanthes linearis* var. *pusilla* Grunov		2											
*Adlafia aquaeductae* (Krasske) Lange-Bertalot		3											
*Caloneis bacillum* (Grunow) Cleve	5		2	6	6						15		15
*Caloneis lancettula* (Schulz) Lange-Bertalot et Witkowski											1		
*Caloneis dubia* Krammer	1												
*Diatoma tenuis* Agardh	3												
*Diatoma vulgaris* Bory		1											
*Eunotia bilunaris* (Ehrenberg) Schaarschmidt		1											
*Eunotia curtagrunowii* Nörpel-Schempp and Lange-Bertalot		15									1		
*Eunotia fallax* A. Cleve	10	6											
*Eunotia paludosa* Grunow	2												
*Eunotia sudetica* f. *minor* Manguin in Bourrelly et Manguin		3											
*Fragilariforma virescens* var. *exigua* (Grunow) M.Poulin		2											
*Gomphonema parvulum* (Kützing) Kützing												1	
*Hantzschia amphioxys* (Ehrenberg) Grunow	2	2											
*Humidophila contenta* (Grunow) R.L.Lowe, Kociolek, J.R.Johansen, Van de Vijver, Lange-Bertalot and Kopalová		15	1										
*Luticola mutica* (Kützing) D.G.Mann		6	1	5									
*Muelleria gibbula* (Cleve) S.A. Spaulding et E.F. Stoermer		4											
*Navicula cincta* (Ehrenberg) Ralfs			3										
*Nitzschia* cf. *ovalis* H.J. Arnott				1									
*Nitzschia palea* (Kützing) W.Smith	4												
*Pinnularia borealis* Ehrenberg		15		4	1						15		
*Pinnularia intermedia* (Lagerstedt) Cleve		2		1								1	
*Pinnularia microstauron* (Ehrenberg) Cleve	1				1							1	
*Pinnularia cf.subcapitata* W. Gregory	7	7	2	1	1				1	2	12		
*Pinnularia* sp.1		1											
*Pinnularia* sp.2				1									
*Pinnularia* sp.3											3		
*Planothidium lanceolatum* (Brébisson ex Kützing) Lange-Bertalot				1									
*Platessa oblongella* (Østrup) C.E. Wetzel, Lange-Bertalot et Ector	1	2											
*Psammothidium ventrale* (Krasske) Bukhtiyarova et Round		4											
*Sellaphora bacillum* (Ehrenberg) D.G.Mann		1											
*Sellaphora mutata (Krasske) Lange-Bertalot*	1	1											
*Sellaphora pupula* (Kützing) Mereschkovsky		1											
*Sellaphora submuralis* (Hustedt) C.E.Wetzel, L.Ector, B.Van de Vijver, Compère et D.G. Mann	1												
*Stauroneis anceps* Ehrenberg				1									
*Staurosirella pinnata* (Ehrenberg) D.M.Williams et Round	3	10											
*Tabularia fasciculata* (C. Agardh) D.M. Williams et Round											1		
Total number	13	22	5	9	4	0	0	0	1	1	7	3	1

Note. The numbers indicate the abundance points.

## Data Availability

Not applicable.

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
