# Peer review of "Diatoms in Volcanic Soils of Mutnovsky and Gorely Volcanoes (Kamchatka Peninsula, Russia)"

_microorganisms, 2021, doi:10.3390/microorganisms9091851_

Round 1
Reviewer 1 Report
The authors aim at uncovering the diatom flora of several sites around two active volcanos on Kamchatka peninsula in the Russian far east. Objectives and Methods are well explained.
Suggest organising Fig 1 differently. Letters indicating sites at red dots in the two aerial photographs are now barely visible. Make the two aerial photos A and B of the two volcanos much bigger in the map of the general vicinity with Petropavlovsk and the two volcanos! Indicate these simply with a clear A and B. Then you can simply refer to A: Gorely and B, Mutnovsky Volcano. Make a small inset showing Kamchatka with a bit of the Russian Far East. No need to show the whole planet. Mark in it a square showing location of more detailed map. In all, the legend of Fig. 1 is not clear.
What about putting in the pictures of Fig. 2 red dots where the actual sample spots were located (if possible) to give the reader an idea of what was sampled. I do note that the GPS geolocations in Table 2 are very precise.
Not clear what the numbers in Table 3 refer to; specimens observed? If so, then the very low numbers put severe limits on any statistics and on the interpretation of diversity. One is wondering if this was the appropriate way to sample the diversity in these scarcely inhabited soils.
In order to help the reader making some ecological appreciations, add to Table 3 information about the sites, their humidity, organic material, pH, salinity, soil type. etc. I know this is partly provided in Table 2 but it is needed here for comparison.
In results authors make statements about ecological groups. I wonder where these categorisations come from. The sources should be defined in the Materials and Methods.
I do not know what “Volcanic destructive soils” are. Regolith?
The discussion could reflect immediately and more clearly on the results gathered in this study, and then compare these with what was found near other volcanos elsewhere, and what the major factors are determining diversity. Also, some remarks about the low numbers of specimens found. Could they be blown in via dust from elsewhere?
Were efforts done to observe living diatoms in the samples by short incubation in the lab?
Minor things
Figure 1. Study area. The red star indicates red star.
Mutnovsky is written in different ways in text and figure legends.
Tabularia fasciculate?
after volcanos eruption – remove the s.
Author Response
Dear reviewer,
We would like to thank the reviewer for careful and thorough reading of this manuscript and for
the thoughtful comments and constructive suggestions, which help to improve the quality of this
manuscript. Our response follows (the reviewer’s comments are in italics and bold).
Reviewer 1
The authors aim at uncovering the diatom flora of several sites around two active volcanos on Kamchatka peninsula in the Russian far east. Objectives and Methods are well explained.
Suggest organising Fig 1 differently. Letters indicating sites at red dots in the two aerial photographs are now barely visible. Make the two aerial photos A and B of the two volcanos much bigger in the map of the general vicinity with Petropavlovsk and the two volcanos! Indicate these simply with a clear A and B. Then you can simply refer to A: Gorely and B, Mutnovsky Volcano. Make a small inset showing Kamchatka with a bit of the Russian Far East. No need to show the whole planet. Mark in it a square showing location of more detailed map. In all, the legend of Fig. 1 is not clear.
We agree, Fig 1 was modified according the reviewer comments.
What about putting in the pictures of Fig. 2 red dots where the actual sample spots were located (if possible) to give the reader an idea of what was sampled. I do note that the GPS geolocations in Table 2 are very precise.
Unfortunately, we have no software and skills for putting in the pictures the actual sample spots.
Not clear what the numbers in Table 3 refer to; specimens observed? If so, then the very low numbers put severe limits on any statistics and on the interpretation of diversity. One is wondering if this was the appropriate way to sample the diversity in these scarcely inhabited soils.
The numbers in Table 3 indicate the abundance points. We added the explanation into the Table 3. The abundance of species was estimated on a coverslip according to the method of R.R. Kabirov. During microscopic observation, five transects were examined: four along the perimeter and one through the center. The abundance of algae was assessed on a 15-point scale on a slide: 1 point, 1–3 diatom valves found on the transect (1–15 on a slide); 2 points, 4–10 diatom valves (4–80 on a slide); 3 points, more than 10 diatom valves (more than 50 on a slide). After observation of five transects, the sum of the abundance points was calculated. Thus, the minimum abundance was 1 point (1–15 on a slide) and the maximum was 15 points (more than 50 on a slide).
In order to help the reader making some ecological appreciations, add to Table 3 information about the sites, their humidity, organic material, pH, salinity, soil type. etc. I know this is partly provided in Table 2 but it is needed here for comparison.
We added the information about the pH and humidity and organic material to Table 2. We were not able to find information about the organic material of studied sites in the literature.
In results authors make statements about ecological groups. I wonder where these categorisations come from. The sources should be defined in the Materials and Methods.
The information about the ecology of the species was find from the relevant literature sources (71-99 in the references list) , such as Ettl, Gärtner, 1995; Kociolek et al., 2003; Lowe, 2003; Stenina, 2009; Antonelli et al., 2016
I do not know what “Volcanic destructive soils” are. Regolith?
It is mountain-tundra volcanic illuvial-humus soils. It is specific soils of Kamchatka peninsula. Corrections was made in the Table 2.
The discussion could reflect immediately and more clearly on the results gathered in this study, and then compare these with what was found near other volcanos elsewhere, and what the major factors are determining diversity. Also, some remarks about the low numbers of specimens found. Could they be blown in via dust from elsewhere?
We corrected the discussion and added the new information according the reviewer comment.
Were efforts done to observe living diatoms in the samples by short incubation in the lab?
We studied the diversity of algae and cyanobacteria in the studied samples, using the dilution method. We found 20 species: Cyanobacteria – 4, Chlorophyta – 14 (Chlorophyceae – 9, Trebouxiophyceae – 5), Streptophyta – 1 и Eustigmatophyceae – 1. But we could not find any diatoms in the samples. Possibly, the abundancy of diatoms in such soils is very low, and it is necessary to use special methods for this group (coverslips).
Minor things
Figure 1. Study area. The red star indicates red star.
Corrected. The red star indicates Petropavlovsk-Kamchatsky city.
Mutnovsky is written in different ways in text and figure legends.
The name of volcano was corrected to “Mutnovsky” everywhere.
Tabularia fasciculate?
Tabularia fasciculate were corrected to Tabularia fasciculata
after volcanos eruption – remove the s.
The mistake was corrected.
Sincerely yours,
Lira Gaysina
Reviewer 2 Report
The article "Diatoms in volcanic soils of Mutnovskiy and Gorely volcanoes
(Kamchatka peninsula, Russia)" is a nice piece of work.
Authors sampled volcanic area of a difficult to access place.
From what regards the protocol, the authors detailled it quite extensively.
My own experience with freshwater diatoms isn't that large, so I may not be that helpful on it, but I may raise two concerns.
One that the authors should address regards the quality of English. Some sentences strongly need to be corrected. The grammar needs to be checked. A sentence like "and initiate the succession during their over-
growth" is odd, for example.
I would also suggest the authors to check the legends below the figures and the tables. For example, in 1A, there is some cyrillic remaining, which might not belong here.
The second regards the taxonomic tools, but for this I guess that the authors might only had access to LM. In general, please keep in mind the risk of having cryptic species. A species like Nitzschia ovalis falls there. This is supposed to be an alkaline (or saline) species, as the authors underlined in the text. There are several small Nitzschia spp. which might prove difficult to identify based on LM, and could be in fact new/different species. If the authors want, they may as well introduce a cf. ovalis name rather. It won't shock me.
SEM pictures would have been a plus obviously. Live LM pictures could have also been used to illustrate the ability of diatoms to live in such place.
I wish the authors the best of luck for the following of the submission process
Author Response
Dear reviewer,
Thanks so much for your feedback and valuable comments! We corrected the MS according your suggestions. Our response follows (the reviewer’s comments are in italics and bold).
Reviewer 2
The article "Diatoms in volcanic soils of Mutnovskiy and Gorely volcanoes
(Kamchatka peninsula, Russia)" is a nice piece of work.
Authors sampled volcanic area of a difficult to access place.
From what regards the protocol, the authors detailled it quite extensively.
My own experience with freshwater diatoms isn't that large, so I may not be that helpful on it, but I may raise two concerns.
One that the authors should address regards the quality of English. Some sentences strongly need to be corrected. The grammar needs to be checked. A sentence like "and initiate the succession during their over-
growth" is odd, for example.
We used mdpi Ehglish correction service.
I would also suggest the authors to check the legends below the figures and the tables. For example, in 1A, there is some cyrillic remaining, which might not belong here.
We deleted the Cyrillic words from the Table 1A.
The second regards the taxonomic tools, but for this I guess that the authors might only had access to LM. In general, please keep in mind the risk of having cryptic species. A species like Nitzschia ovalis falls there. This is supposed to be an alkaline (or saline) species, as the authors underlined in the text. There are several small Nitzschia spp. which might prove difficult to identify based on LM, and could be in fact new/different species. If the authors want, they may as well introduce a cf. ovalis name rather. It won't shock me.
We corrected Nitzschia ovalis to Nitzschia cf.ovalis.
SEM pictures would have been a plus obviously. Live LM pictures could have also been used to illustrate the ability of diatoms to live in such place.
Unfortunately, we have no access to SEM microscope now, but we will solve this matter in the future. We studied the diversity of algae and cyanobacteria in the studied samples, using the dilution method. We found 20 species: Cyanobacteria – 4, Chlorophyta – 14 (Chlorophyceae – 9, Trebouxiophyceae – 5), Streptophyta – 1 и Eustigmatophyceae – 1. But we could not find any diatoms in the samples. Possibly, the abundancy of diatoms in such soils is very low, and it is necessary to use special methods for this group (coverslips).
I wish the authors the best of luck for the following of the submission process
Thank you very much for the nice wishes!
Sincerely yours,
Lira Gaysina